# What do Deck Chairs and Sun Hats Have in Common?
# Uncovering Shared Properties in Large Concept Vocabularies

**Amit Gajbhiye[1], Zied Bouraoui[2], Na Li[3], Usashi Chatterjee[1],**
**Luis Espinosa Anke[1,4], Steven Schockaert[1]**

[1] CardiffNLP, Cardiff University, UK     [2] CRIL CNRS & University of Artois, France
[3] University of Shanghai for Science and Technology, China     [4] AMPLYFI, UK
{gajbhiyea,chatterjee,espinosa-ankel,schockaerts1}@cardiff.ac.uk
zied.bouraoui@cril.fr   li_na@usst.edu.cn

## Abstract

Concepts play a central role in many applications. This includes settings where concepts have to be modelled in the absence of sentence context. Previous work has therefore focused on distilling decontextualised concept embeddings from language models. But concepts can be modelled from different perspectives, whereas concept embeddings typically mostly capture taxonomic structure. To address this issue, we propose a strategy for identifying what different concepts, from a potentially large concept vocabulary, have in common with others. We then represent concepts in terms of the properties they share with the other concepts. To demonstrate the practical usefulness of this way of modelling concepts, we consider the task of ultra-fine entity typing, which is a challenging multi-label classification problem. We show that by augmenting the label set with shared properties, we can improve the performance of the state-of-the-art models for this task.[1]

## 1 Introduction

Various applications rely on knowledge about the meaning of concepts, which is typically encoded in the form of embeddings. For instance, pre-trained concept embeddings are used to provide prior knowledge about the labels in few-shot and zero-shot learning tasks (Socher et al., 2013; Ma et al., 2016; Xing et al., 2019; Yan et al., 2022; Xiong et al., 2019; Hou et al., 2020; Li et al., 2020; Yan et al., 2021). Concept embeddings also play a central role in the area of knowledge management, for instance for taxonomy learning (Vedula et al., 2018; Malandri et al., 2021), knowledge graph alignment (Trisedya et al., 2019), ontology alignment (Kolyvakis et al., 2018) and ontology completion (Li et al., 2019). In such applications, Language Models (LMs) such as BERT (Devlin et al., 2019) cannot be used directly, since we need

concept embeddings which do not depend on sentence context. For this reason, several authors have proposed strategies for obtaining decontextualised (or static) concept embeddings from LMs (Ethayarajh, 2019; Bommasani et al., 2020; Vulić et al., 2020; Li et al., 2021; Vulić et al., 2021; Liu et al., 2021; Gajbhiye et al., 2022).

Decontextualised concept embeddings tend to primarily reflect basic taxonomic structure, e.g. capturing the fact that televisions are electronic devices and that salmon are fish. However, applications often rely on different facets of meaning. In computer vision applications, we want concept embeddings to reflect visual features; e.g. they should capture the fact that chess boards, televisions and books have something in common (i.e. being rectangular). When modelling scene graphs (Qi et al., 2019), we may rather want embeddings that capture which objects are often found together in the same visual scene. For instance, concept embeddings should then reflect the fact that televisions, sofas and rugs have something in common (i.e. that they are typically found in the living room). When modelling food ontologies, concept embeddings should perhaps capture the fact that salmon and walnuts share the property of being rich in Omega-3. In principle, if sufficient training data is available, we could learn application-specific concept embeddings by fine-tuning a language model. However, the role of concept embeddings is often precisely to address the scarcity of application-specific training data. We, therefore focus on strategies that use pre-trained models in an unsupervised way.

Concept embeddings can be viewed as compact representations of similarity metrics. However, what matters in the aforementioned applications is often less about modelling similarity than about modelling what different concepts have in common. This is an important distinction because we can list the properties that a concept satisfies in an application-independent way, which is not possi-

---

[1]Our datasets and evaluation scripts are available at `https://github.com/amitgajbhiye/concept_commonality`

| Property | Concepts |
|---|---|
| used for staying connected | telephone number, google, area code, user name, facebook, land line, email, ~~call center~~, ~~web server~~, ~~link farm~~, ~~name server~~ |
| used for beach activities | deck chair, beach volleyball, beach ball, sun hat, ~~sun dog~~ ... |
| a round shape | bubble, small ball, ball bearing, sphere, disc, round top, centre circle, wheel, tin can, ~~cone~~, ~~cylinder~~, ~~oval~~, ~~tube~~ |
| located in cities | parking lot, high street, suburbs, supermarket, mall, food court, piazza, city hall, cinema, ~~general store~~ |
| used for communication | address book, twitter, voice message, instant message, voice mail ~~letter bomb, listening post, mail bomb, ring tone, street address, wireless operator~~ |
| accommodation options for camping | beer tent, man cave, rock shelter, holiday camp, camp, mobile home, ~~guest room, inn, resort, trading post, cabin, cabin boy, hunting lodge, lodge, mess hall~~ |
| accounting | expense account, transaction, register, business intelligence, finance, business record, income statement, ~~backup, census, consumer credit, estimate, evaluation, listing, maintenance, market research~~ |
| tv series genres | detective, soap opera, reality show, crime, mystery, cartoon ~~blockbuster, hero, radio drama, season finale, series finale~~ |
| aches | fever, strain, head cold, battle fatigue, sore, pain, hurt, fatigue ~~bow shock, cold, dry eye, hunger, obesity, sting, stretch mark~~ |
| adulthood | middle age, thirty, age group, womanhood, drinking age, aged, careers, elder, generation, maturity, ~~autumn, bachelor, birth, birthplace, childhood, decade, era~~ |
| advertisements | 'slogan', 'brand image', 'publicity', 'advertising', 'product placement', 'banner ad', 'advertiser', 'advertisement', ~~'advance', 'propaganda', 'ambush marketing', 'dot product', 'fan mail', 'fan service'~~ |
| baby carriers | basket, baby seat, basket case, car seat, carrier, ~~baby book, baby bottle, body bag~~ |

Table 1: Examples of properties found using the bi-encoder model. In red and crossed out are properties initially predicted, but discarded by the DeBERTa-based filtering.

ble for modelling similarity. Based on this view, we propose the following strategy: we represent concepts by explicitly capturing the properties they satisfy. To identify the properties that might be relevant in a given application, we look for those that describe what different concepts from the given vocabulary have in common.

To implement this strategy, we need an efficient mechanism for identifying these commonalities. The vocabulary is often too large to directly use Large Language Models (LLMs) to solve this task. Instead, we rely on a bi-encoder architecture to efficiently retrieve the properties that are satisfied by each concept (Gajbhiye et al., 2022). Once the initial set of properties has been retrieved, we use a fine-tuned DeBERTa model (He et al., 2021) to filter these properties. Table 1 shows examples of properties that were thus identified, along with the corresponding concepts. Note how some of these commonalities are unlikely to be captured by standard concept embeddings (e.g. linking *telephone number* and *facebook* in the first example).

After summarising the related work in Section 2, we describe our strategy for uncovering shared properties in Section 3. Our subsequent evaluation in Section 4 focuses on two tasks. First, we carry out an intrinsic evaluation to show the effectiveness of the proposed filtering strategy. Second, we

demonstrate the practical usefulness of uncovering shared properties on the downstream task of ultra-fine entity typing.

## 2 Related Work

The task of predicting commonsense properties has been studied by several authors (Rubinstein et al., 2015; Forbes et al., 2019; Gajbhiye et al., 2022; Apidianaki and Garí Soler, 2021; Bosselut et al., 2019). Modelling the commonalities between concepts, or conversely, identifying outliers, has also been previously considered. For instance, outlier detection has been used for intrinsic evaluation of word embeddings (Camacho-Collados and Navigli, 2016; Blair et al., 2017; Brink Andersen et al., 2020). However such benchmarks are focused on taxonomic categories. Moreover, we focus on finding commonalities in large vocabularies, containing perhaps tens of thousands of concepts. Another related task is entity set expansion (Pantel et al., 2009; Zhang et al., 2020; Shen et al., 2020). Given an initial set of entities, e.g. {*France, Germany, Italy*}, this task requires selecting other entities that have the same properties (e.g. being European countries). However, the focus is usually on named entities, whereas we focus on concepts.

## 3 Identifying Shared Properties

Let $\mathcal{V}$ be the considered vocabulary of concepts. Our aim is to associate each concept $c \in \mathcal{V}$ with a set of properties. To find suitable properties, we first retrieve a set of candidate properties for each concept in $\mathcal{V}$ using an efficient bi-encoder model. Subsequently, we verify the retrieved properties with a joint encoder.

**Retrieving Candidate Properties** For each concept $c \in \mathcal{V}$, we want to find a set of properties that are likely to be satisfied by that concept. The vocabulary $\mathcal{V}$ may contain tens of thousands of concepts, which makes it expensive to use an LLM for this purpose. We instead rely on the bi-encoder model from Gajbhiye et al. (2022). The idea is to fine-tune two BERT models: one for learning concept embeddings ($\phi_{\mathsf{con}}$) and one for learning property embeddings ($\phi_{\mathsf{prop}}$). The probability that concept $c$ has property $p$ is then estimated as $\sigma\left(\phi_{\mathsf{con}}(c) \cdot \phi_{\mathsf{prop}}(p)\right)$, with $\sigma$ the sigmoid function. To train this model, we need examples of concepts and the properties they satisfy. The only large-scale knowledge base that contains such training examples is ConceptNet, which is unfortunately rather noisy and imbalanced. Gajbhiye et al. (2022) therefore trained their model on a large set of (hyponym,hypernym) pairs from Microsoft Concept Graph (Ji et al., 2019), together with examples from GenericsKB (Bhakthavatsalam et al., 2020).

Since the performance of the bi-encoder heavily depends on the training data, and existing training sets are sub-optimal, we created a training set of 109K (concept,property) pairs using ChatGPT[2]. Simply asking ChatGPT to enumerate the properties of some concept tends to result in verbose explanations describing overly specific properties, which are less helpful for identifying commonalities. Therefore, we used a prompt which specifically asked ChatGPT to identify properties that are shared by several concepts. We also experimented with a training set we derived from ConceptNet 5.5. Specifically, we converted instances of the relations *IsA*, *PartOf*, *LocatedAt*, *UsedFor* and *HasProperty* into a set of 63,872 (concept,property) pairs. The full details can be found in the appendix.

**Selecting Properties** To associate properties with concepts, we first need to determine a set $\mathcal{P}$ of properties of interest. For our experiments, we let

| Model | Pre-training | F1 |
|---|---|---|
| BERT-large bi-enc | - | 36.6 |
| BERT-large bi-enc | (Gajbhiye et al., 2022) | 49.3 |
| BERT-large bi-enc | ConceptNet | 54.0 |
| BERT-large bi-enc | ChatGPT | 50.1 |
| BERT-large bi-enc | ConceptNet+ChatGPT | 55.4 |
| BERT-large joint | - | 51.8 |
| RoBERTa-large joint | - | 58.8 |
| DeBERTa-large joint | - | **65.9** |
| BERT-large joint | ConceptNet | 55.4 |
| RoBERTa-large joint | ConceptNet | 60.3 |
| DeBERTa-large joint | ConceptNet | 65.7 |
| RoBERTa-large NLI | - | 57.7 |
| RoBERTa-large NLI | WANLI | 57.2 |
| RoBERTa-large NLI | WANLI + ConceptNet | 57.6 |

Table 2: Results on the McRae property split dataset.

$\mathcal{P}$ contain every property that appears at least twice in the training data for the bi-encoder. Then for a given concept $c$, we use maximum inner-product search (MIPS) to efficiently find the 50 properties $p$ from $\mathcal{P}$ for which the dot product $\phi_{\mathsf{con}}(c) \cdot \phi_{\mathsf{prop}}(p)$ is maximal. To make a hard selection of which properties are satisfied and which ones are not, we rely on a joint encoder. Such models are typically more accurate than a bi-encoder but cannot be used for retrieval. Specifically, we use a masked language model with the following prompt:

$$\textit{Can c be described as p? [MASK].} \qquad (1)$$

We train a linear classifier to predict whether $c$ has the property $p$ from the final layer embedding of the [MASK] token. As the masked language model, we use a DeBERTa-v3-large model. To train the classifier and fine-tune the language model, we used the extended McRae dataset from Forbes et al. (2019) and the augmented version of CSLB[3] introduced by Misra et al. (2022). Both datasets have true negatives and are focused on commonsense properties, which we found important in initial experiments. Finally, after obtaining the properties for every concept in $\mathcal{C}$, we remove those properties that were only found for a single concept.

## 4 Experiments

**Predicting Commonsense Properties** As an intrinsic evaluation of our model for predicting properties, we use the property split of the McRae dataset that was introduced by Gajbhiye et al. (2022). This benchmark involves classifying (concept,property) pairs as valid or not. It is particularly

---

[2] https://chat.openai.com

[3] https://cslb.psychol.cam.ac.uk/propnorms

| Representation | F1 |
|---|---|
| Base model (Li et al., 2023a) | 49.2 |
| ConCN clusters (Li et al., 2023a) | 50.4 |
| Properties (ChatGPT) | 50.4 |
| Properties (ConceptNet) | 50.9 |
| Properties (ChatGPT + CN) | 50.9 |
| Bi-enc clusters (ChatGPT + CN) | 50.6 |
| ConCN clusters + properties (ChatGPT + CN) | 50.9 |
| Bi-enc clusters + properties (ChatGPT + CN) | **51.1** |

Table 3: Results for ultra-fine entity typing, using a BERT-base entity encoder with augmented label sets.

| Representation | F1 |
|---|---|
| Base model (Li et al., 2023a) | 49.8 |
| ConCN clusters (Li et al., 2023a) | 51.9 |
| Bi-enc clusters + properties (ChatGPT + CN) | **52.2** |

Table 4: Results for ultra-fine entity typing, using a BERT-large entity encoder and after applying the post-processing strategies from Li et al. (2023a).

challenging because there is no overlap between the properties in the training and test splits.

The results for the bi-encoder are shown in the top part of Table 2. As can be seen, the training sets obtained from ChatGPT and from ConceptNet both outperform the dataset that was used by Gajbhiye et al. (2022). The best results were obtained by combining the ChatGPT and ConceptNet training sets. In the middle part of Table 2, we evaluate different variants of the joint encoder. We compare three masked language models: BERT-large, RoBERTa-large (Liu et al., 2019) and DeBERTa-v3-large (He et al., 2021). In each case, we consider one variant where the models are directly fine-tuned on the McRae training set and one where the models are first pre-trained on ConceptNet (i.e. the dataset we used for training the bi-encoder). As can be seen, the best results are obtained using DeBERTa without pre-training on ConceptNet. One disadvantage of the ConceptNet dataset is that it does not contain true negatives, i.e. the negative training examples are obtained by randomly corrupting positive examples. We also experimented with an NLI formulation. To this end, we used a premise of the form "*the concept is c*" and a hypothesis of the form "*the concept can be described as p*". The results are shown in the bottom part of Table 2 for three variants with RoBERTa-large: one without pre-training, one where we pre-train on WANLI (Liu et al., 2022), and one where we first pre-train on WANLI and then continue training on ConceptNet. As can be seen, this NLI based formulation was less successful.

**Ultra-Fine Entity Typing**   To evaluate the usefulness of identifying shared properties in a downstream task, we consider the task of ultra-fine entity typing (Choi et al., 2018). Given a sentence in which an entity is highlighted, this task consists in assigning semantic types to that entity. It is formu-

lated as a multi-label classification problem, where there are typically several labels that apply to a given entity. The task is challenging due to the fact that a large set of more than 10,000 candidate labels is used. Moreover, most of the training data comes from distant supervision signals, and many labels are not covered by the training data at all. This makes it important to incorporate prior knowledge about the meaning of the labels. Li et al. (2023a) recently achieved state-of-the-art results with a simple clustering based strategy. They first cluster all the labels based on a pre-trained embedding at different levels of granularity. Each cluster is then treated as an additional label. For instance, if the label $l$ appears in clusters $c_1, ..., c_k$ then whenever the label $l$ appears in a training example, the labels $c_1, ..., c_k$ are added as well. This simple label augmentation strategy was found to lead to substantial performance gains, as long as high-quality concept embeddings were used. Their best results were achieved using the ConCN concept embeddings from Li et al. (2023b).

In this experiment, we use the shared properties that we uncovered as additional labels, in the same way that Li et al. (2023b) used clusters. For instance, if the property *found in the wild* was found for *elephant*, then whenever a training example is labelled with *elephant* we add the label *found in the wild*. To identify the shared properties, we use the same bi-encoders that we used for the experiment in Table 2. We filter the properties using the DeBERTa-large model that was fine-tuned on McRae and CSLB.

The results are summarised in Table 3, for the case where a BERT-base entity encoder is used. The base model, without any additional labels, is the DenoiseFET model from Pan et al. (2022). When adding the clusters of ConCN embeddings, Li et al. (2023a) were able to increase the F1 score from 49.2 to 50.4. The configurations where we instead use the shared properties are reported as *Properties (X)*, with $X$ the training set that was

used for the bi-encoder. As can be seen, with the bi-encoder trained on ConceptNet, we achieve an F1 score of 50.9, which clearly shows the usefulness of the shared properties. The model that was trained on both ConceptNet and the ChatGPT examples achieves the same result. Next, we tested whether the shared properties and the clusters used by Li et al. (2023b) might have complementary strengths (*ConCN + properties*). We also considered a variant where we instead obtained clusters from the concept encoder of the bi-encoder model (*Bi-enc + properties*). For both variants, we used the bi-encoder that was trained on Chat-GPT and ConceptNet. Adding the clusters from the bi-encoder leads to a further improvement to 51.1. With the ConCN clusters, the result remains unchanged. When only the clusters from the bi-encoder are used, we achieve an F1 score of 50.6. Finally, Table 4 shows the result of our best configuration when using a BERT-large entity encoder, and when applying the post-processing techniques proposed by Li et al. (2023a). As can be seen, our model surpasses their state-of-the-art result.

## 5 Conclusions

Concept embeddings are often used to provide prior knowledge in applications such as (multi-label) few-shot learning. We have proposed an alternative to the use of embeddings for such applications, where each concept is instead represented in terms of the properties it satisfies. Our motivation comes from the observation that concept embeddings tend to primarily capture taxonomic relatedness, which is not always sufficient. We first use a bi-encoder to efficiently retrieve candidate properties for each concept, and then use a joint encoder to decide which properties are satisfied. The resulting property assignments have allowed us to improve the state-of-the-art in ultra-fine entity typing.

## Limitations

A key advantage of concept embeddings is that they can straightforwardly be used as input features to neural network models. Our representations based on shared properties may be less convenient to work with in cases where concept representations have to be manipulated by a neural network. In our experiments on ultra-fine entity typing, this was not necessary as the shared properties were merely used to augment the training labels. To adapt our approach to different kinds of tasks, further work

may be needed. For instance, it may be possible to compactly encode the shared properties as dense vectors, e.g. by using partially disentangled representations to separate different aspects of meaning.

**Acknowledgments**   This work was supported by EPSRC grant EP/V025961/1, ANR-22-CE23-0002 ERIANA and by HPC resources from GENCI-IDRIS (Grant 2023-[AD011013338R1]).

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

## A Collecting Concept-Property Pairs using ChatGPT

To obtain training data for the bi-encoder model using ChatGPT, we used the following prompt:

> *I am interested in knowing which properties are satisfied by different concepts. I am specifically interested in properties, such as being green, being round, being located in the kitchen or being used for preparing food, rather than in hypernyms. For instance, some examples of what I'm looking for are: **1. Sunflower, daffodil, banana are yellow 2. Guitar, banjo, mandolin are played by strumming or plucking strings 3. Pillow, blanket, comforter are soft and provide comfort 4. Car, scooter, train have wheels 5. Tree, log, paper are made of wood 6. Study, bathroom, kitchen are located in house**. Please provide me with a list of 50 such examples.*

Providing a number of examples as part of the prompt helped to ensure that all answers followed the same format, specifying one property and three concepts that satisfy it. One difficulty we faced is that by repeating the same prompt, after a while the model started mostly generating duplicates of concept-property pairs that we already collected. To address this, regularly changed the examples in the prompt. Specifically, each question asked for 50 answers, and we repeated a question with the same prompt around 10 times, after which we changed the examples, to prevent the model from generating

too many duplicates. This process allowed us to obtain a dataset with 109K unique (concept,property) pairs.

## B Collecting Concept-Property Pairs from ConceptNet

We compiled a training set of concept-property pairs for training the bi-encoder model from ConceptNet 5.5. ConceptNet contains a broad range of relations, many of which are not relevant for our purposes (e.g. relations about events). Following Bhakthavatsalam et al. (2020), we considered the ConceptNet relations *IsA*, *PartOf*, *LocatedAt* and *UsedFor*. In addition, we also use the *HasProperty* relation, which is clearly relevant. To convert ConceptNet triples into concept-property pairs, we need to choose a verbalisation of the relationships. We explain this process by providing an example for each relation type: the ConceptNet triple (*apartments*, *PartOf*, *apartment buildings*) is mapped to the concept-property pair (*apartments*, *part of apartment buildings*); the triple (*seaplane*, *IsA*, *airplane*) is mapped to (*seaplane*, *airplane*); the triple (*airplane*, *UsedFor*, *travelling*) is mapped to (*airplane*, *used for travelling*); the triple (*airplane*, *AtLocation*, *the sky*) is mapped to (*airplane*, *located in the sky*).

## C Additional Details

**Bi-encoder Model** The bi-encoders are trained using binary cross-entropy. We train the model for 100 epochs, using early stopping with a patience of 10 (using a 10% held-out portion of the training data for validation). We optimise the model parameters using the AdamW optimizer (Loshchilov and Hutter, 2019) with a learning rate of $2e-6$. We employ a batch size of 8 and $L_2$ weight decay of 0.1.

**Joint encoder** The joint encoders are trained using binary cross-entropy. We train the model to a maximum of 100 epochs with an early stopping patience of 5. A learning rate of $1e-5$ and batch size of 32 is used to train the model. Further, we use the $L_2$ weight decay of 0.01.

At the time of verifying the properties using the joint encoder, to avoid introducing too much noise, a property $p$ is only assigned to a concept $c$ if the confidence of our DeBERTa classifier is at least $\lambda$, for a hyperparameter $\lambda$. We consider values of $\lambda \in \{0.5, 0.75, 0.9\}$. Based on the validation split, we found that $\lambda = 0.75$ gave the best results.

**Ultra-fine entity typing**  To train the entity encoders for the experiments on ultra-fine entity typing, we follow the methodology of Li et al. (2023a). In particular, to obtain clusters of concept embeddings, we use affinity propagation, where we select the preference value from $\{0.5, 0.6, 0.7, 0.8, 0.9\}$ based on the validation split. The entity encoder is trained using the soft prompt based approach from Pan et al. (2022).

## D   Qualitative Analysis

Table 5 shows some of the properties found for the vocabulary from the McRae dataset. For this experiment, we used a version of the DeBERTa model that was only fine-tuned on CSLB. On each row, we compare a predicted property with the most similar property from the ground truth (where similarity was measured in terms of the Jaccard overlap of the corresponding concept sets). One observation is that the properties we uncover are often more specific than those in the ground truth. For instance, *commonly played in orchestra* is clearly more specific than *used for music*. What is also evident from these examples is that our strategy is precision-oriented. For a given property, the set of concepts that are predicted to have that property are mostly correct, but sometimes some clearly relevant concepts are missing. For instance, for *used for cooking food*, concepts such as *blender*, *strainer* and *grater*, among others, are clearly also relevant. Finally, while many of the properties that are identified are taxonomic, there are also several non-taxonomic properties, such as *used for cooking food* and *part of firearms*.

Our motivation was based on the idea that concept embeddings primarily capture taxonomic properties. To test this idea, Table 6 shows the nearest neighbours of the concept embeddings produced by our bi-encoder. The table in particular shows the neighbours of concepts that also appear in Table 6. Compared to the properties that are found in that table, we can see that the neighbours in Table 6 indeed mostly reflect taxonomic similarity. For instance, the top neighbours of *dining table* are different types of tables.

| Predicted property | McRae property |
|---|---|
| *weapons of war*: missile, crossbow, pistol, bullet, sword, shotgun, bazooka, spear, revolver, grenade, rifle, cannon, bayonet, bomb, gun, dagger | *used for killing*: bayonet, revolver, harpoon, crossbow, pistol, bullet, grenade, shotgun, sword, bazooka, rattlesnake, rifle, machete, bomb, catapult, missile, spear, cannon, gun, dagger |
| *used for cooking food*: oven, pan, spatula, skillet, stove, toaster, pot, microwave | *used for cooking*: lemon, cherry, fork, stove, toaster, blender, strainer, mixer, oven, spatula, ladle, clock, pot, microwave, bowl, grater, pan, colander, knife, spoon, tongs, kettle, olive, spinach, skillet, apron |
| *commonly played in orchestras*: trombone, cello, violin, saxophone, trumpet, clarinet, tuba, flute | *used for music*: piano, banjo, trombone, cello, cell phone, harmonica, bagpipe, radio, clarinet, keyboard, tuba, whistle, accordion, rattle, laptop, stereo, trumpet, guitar, harp, drum, violin, saxophone, harpsichord, flute |
| *part of firearms*: muzzle, shotgun, bullet, pistol, bazooka, rifle, revolver | *used for killing*: bayonet, revolver, harpoon, crossbow, pistol, bullet, grenade, shotgun, sword, bazooka, rattlesnake, rifle, machete, bomb, catapult, missile, spear, cannon, gun, dagger |
| *has wheels*: cart, wagon, limousine, taxi, tricycle, bicycle, car, truck, buggy, scooter, trolley, van, motorcycle | *used for transportation*: sailboat, wagon, ambulance, buggy, helicopter, helmet, scooter, motorcycle, yacht, horse, bicycle, train, ship, trailer, canoe, trolley, bus, jet, sled, rocket, pony, limousine, sleigh, truck, surfboard, submarine, saddle, shoes, unicycle, airplane, slippers, bridge, boat, jeep, cart, tricycle, escalator, taxi, car, camel, subway, van, skateboard, elevator, bike, wheel, tractor, raft |
| *mammals commonly found in forests*: rabbit, hare, caribou, coyote, squirrel, chipmunk, bear, bison, groundhog, skunk, moose, deer, fox, raccoon, cougar, beaver, elk | *an animal*: flea, flamingo, python, toad, penguin, rattlesnake, cockroach, iguana, fawn, zebra, ox, crow, pig, grasshopper, gorilla, dove, shrimp, dolphin, woodpecker, buffalo, rat, bison, deer, cat, swan, ostrich, chipmunk, platypus, tortoise, beaver, lion, caribou, butterfly, salmon, moose, clam, seagull, moth, snail, squirrel, housefly, cheetah, walrus, octopus, dog, hornet, mouse, coyote, spider, eagle, turtle, porcupine, giraffe, alligator, donkey, horse, pony, seal, groundhog, caterpillar, raccoon, cougar, elk, hare, tuna, otter, wasp, panther, bear, camel, owl, falcon, fox, frog, whale, hyena, calf, goat, duck, lobster, rabbit, elephant, hamster, goose, pigeon, squid, leopard, chicken, salamander, tiger, crab, hawk, peacock, turkey, sheep, chimp, gopher, bird, eel, crocodile, trout, rooster, cow, skunk, bull, lamb |

Table 5: Comparison of shared properties which are uncovered by our method, for the vocabulary of the McRae dataset, with the properties included in the ground truth of that dataset.

| Concept | Nearest Neighbors |
|---|---|
| telephone number | telephone number, phone number, address, web address, house number, street address |
| deck chair | deck chair, beach chair, swing, lounge, tree house, camp bed, pleasure boat |
| dining table | dining table, dinner table, table, coffee table, dining room, counter, desk |
| bubble | bubble, air bubble, bubble sort, balloon, tube, speech balloon, small ball |
| parking lot | parking lot, car park, parking garage, parking space, bus lane, alley, bicycle lane |

Table 6: Examples of nearest neighbours, in terms of cosine similarity between the embeddings obtained by the concept encoder of the model trained using ConceptNet and ChatGPT. The considered vocabulary is that of the ultra-fine entity typing dataset.