# OpenReview forum: "What do Deck Chairs and Sun Hats Have in Common? Uncovering Shared Properties in Large Concept Vocabularies"
_EMNLP/2023/Conference — EMNLP 2023 Main_

### Official Review · Reviewer_zfkC · 2023-08-01

**Soundness:** 3

**Excitement:**

3: Ambivalent: It has merits (e.g., it reports state-of-the-art results, the idea is nice), but there are key weaknesses (e.g., it describes incremental work), and it can significantly benefit from another round of revision. However, I won't object to accepting it if my co-reviewers champion it.

**Paper Topic And Main Contributions:**

The paper describes an approach for identifying properties shared by concepts in a large vocabulary.
It is based on a bi-encoder for retrieving properties satisfied by each concept and a joint encoder (fine-tuned DeBERTa) for deciding which properties are satisfied.
Performance of different configurations is computed in the "McRae" dataset; and when used in the downstream task of ultra-fine entity typing.

**Questions For The Authors:**

Question A: Would it make sense to refer to Analogy Solving as a related task?

Question B: To what extent would Knowledge Graph embeddings be useful for this task?

Question C: Why not training in the same datasets as Gajbhiye et al. (2022)?

Question D: Would (Open)Cyc be a less noisy alternative to ConceptNet?

Question E: Did you check whether the training data, especially that generated by ChatGPT, did not contain examples in the test?

Question F: ChatGPT was used for generating data, but, out of curiosity, did the authors test whether it could perform the task?

**Reasons To Accept:**

Well-motivated.

Some interesting experiments.

**Reasons To Reject:**

The paper is not self-contained. Should perhaps be a long paper?
Important information is either not present (e.g., description of datasets like MCG, McRae, WANLI, including examples) or only in the Appendix (e.g., creation of the training set, ChatGPT prompt), which is only referred in a footnote, not mentioning the number. The latter make it harder to follow the paper.

**Reproducibility:**

2: Would be hard pressed to reproduce the results. The contribution depends on data that are simply not available outside the author's institution or consortium; not enough details are provided.

**Reviewer Confidence:**

3: Pretty sure, but there's a chance I missed something. Although I have a good feel for this area in general, I did not carefully check the paper's details, e.g., the math, experimental design, or novelty.

**Typos Grammar Style And Presentation Improvements:**

The paper never mentions the language it is on / applies to (English).

The abstract is too high-level: goal, experiments and contributions could be more explicit.

The Introduction does not describe the structure of the paper.

Table 2: not clear where / how the models with the pretraining column empty were pretrained.

line 234: consists in -> consists of

---

> ### Author Rebuttal · Authors · 2023-08-25
>
> We thank the reviewer for their careful reading of our paper.
>
> We will add a table to the appendix, with statistics and examples for each of the considered datasets. It is common to provide such information in the appendix, rather than the main paper, especially since the datasets involved are relatively well-known. Note that the claim that the appendix is not referred to in the text is incorrect (see footnote 2 on page 3).
>
> Question A: While some analogies can be solved by identifying shared properties, the task of analogy solving is mostly about measuring degrees of relational similarity. Moreover, our framework is specifically designed for cases where we have to identify commonalities among a large set of concepts (meaning that the set of concepts cannot be explicitly listed as part of a language model prompt). Analogy solving would thus not be the most natural application.
>
> Question B: It is not clear to us how knowledge graph embeddings could be used for our setting.
>
> Question C: A comparison with the pre-training dataset from Gajbhiye et al. (2022) is made in Table 2, where we can see that our pre-training datasets perform better.
>
> Question D: We have used ConceptNet because it is a well-known and often used dataset in NLP.  Our understanding of OpenCYC is that it is mostly focused on taxonomic relations, and would therefore be less suitable.
>
> Question E: Note that this point only applies to the analysis in Table 2. We found that there is indeed a small amount of overlap between the pre-training datasets and the McRae test set: around 4% of the test set is present in the largest pre-training dataset (i.e. ConceptNet + ChatGPT). However, we do not consider this overlap to be problematic, since the pre-training data was collected independent of the McRae dataset (i.e. the fact that assertions from the McRae test set appear in the pre-training dataset is simply a reflection of the fact that it contains a comprehensive set of commonsense properties). Nonetheless, for completeness, we have now also carried out an evaluation of the bi-encoder models after removing the items from the McRae test set from the pre-training datasets. After removing these items, the resulting performance stays very close to what we report in Table 2:
>
> Bi-encoder with ConceptNet: 53.78,
> Bi-encoder with ChatGPT: 50.37,
> Bi-encoder with ConceptNet + ChatGPT: 54.97
>
> Question F: Using ChatGPT or any other LLM for this task would not be straightforward given the huge space of concepts we deal with in this paper. We would need a model with a context window that is large enough to enumerate the full vocabulary (i.e. around 10000 concepts in the case of the UFET dataset). Moreover, while models with large context windows now exist, based on our experience, GPT-4 already struggles with manipulating much smaller lists of concepts. For instance, asking GPT-4 to rank a list of around 100 concepts according to some criteria already leads to various issues (e.g. missing and duplicated concepts).
>
> We will address the suggestions that were provided in the section on “presentation improvements”.

---

### Official Review · Reviewer_k24T · 2023-08-05

**Soundness:** 3

**Excitement:**

4: Strong: This paper deepens the understanding of some phenomenon or lowers the barriers to an existing research direction.

**Paper Topic And Main Contributions:**

The paper is devoted to a strategy for identifying what different concepts, from a potentially large concept vocabulary, have in common with others. They want to associate each concept with a set of properties and use a bi-encoder model. Then they verify the retrieved properties with a joint encoder.

**Reasons To Accept:**

The paper tackles an interesting and a relevant problem and considers concepts and properties from a different angle (not new but quite uncommon). The authors compute concept embeddings based on properties and perform both intrinsic and extrinsic evaluation.

**Reasons To Reject:**

I am leaning towards accepting the paper, here are some things how the paper could be improved for better understanding of the task and better presentation of the research to the Reader.

 First, it is not clear how the choice on pertaining datasets was made. Why for bi-encoders the authors experiment with both CN and ChatGPT data, then for joint encoders the authors use ConceptNet only but for the ET experiments they use ChatGPT data again.

It is also not clear, whether concept embeddings can be computed for any model (bi-enc or joint) and whether bi-encoder from the first part is the same for the Entity Typing experiments. Otherwise, which embeddings exactly are used for the ET experiments?

**Reproducibility:**

3: Could reproduce the results with some difficulty. The settings of parameters are underspecified or subjectively determined; the training/evaluation data are not widely available.

**Reviewer Confidence:**

4: Quite sure. I tried to check the important points carefully. It's unlikely, though conceivable, that I missed something that should affect my ratings.

**Typos Grammar Style And Presentation Improvements:**

I would suggest writing a list of contribution in the Intro section.

I would also recommend to compute stdev for the Entity Typing experiments as numbers are very similar and it is not clear how different 50.9 and 51.1 are and how much we could benefit from your approach.

---

> ### Author Rebuttal · Authors · 2023-08-25
>
> We thank the reviewer for their careful reading of our paper.
>
> Regarding the choice of pre-training datasets, for the evaluation of the bi-encoder (Table 2) and the UFET model (Table 3), we have presented a full comparison of the three considered pre-training datasets: ConceptNet, ChatGPT, and the combination of both. For the best-performing joint encoder (i.e. the DeBERTa model), the main message is that pre-training does not help. This is true for all pre-training datasets, but we did not include the full analysis as this was essentially a negative result.
>
> The joint encoder cannot be used to generate embeddings. We use the bi-encoder to generate embeddings. The embeddings are used to select candidate properties. These candidate properties are finally filtered using the joint encoder.
>
> The bi-encoder and joint encoders that were used for the ultra-fine entity typing experiments are indeed the pre-trained models. The bi-encoders are identical to the ones that were used in the first part; the joint encoder for the UFET experiments was fine-tuned both on McRae and CSLB.
>
> We will clarify these points and address the two suggestions from the section on “presentation improvements”.

---

### Official Review · Reviewer_vHKK · 2023-08-07

**Soundness:** 3

**Excitement:**

4: Strong: This paper deepens the understanding of some phenomenon or lowers the barriers to an existing research direction.

**Paper Topic And Main Contributions:**

The work introduces a strategy for identifying what different concepts have in common by analysing their shared properties. Specifically, two BERT models are fine-tuned to learn concepts and properties embedding separately. The authors employ ChatGPT to enumerate concept properties, and then a selection of data is led by finding the 50 properties using a linear classifier.

**Reasons To Accept:**

The idea is exciting and in time with the recent advancements in the Natural Language Processing (NLP) field.

**Reasons To Reject:**

Employing ChatGPT may identify a weakness in generating concept properties (it suffers from the hallucination problem).

The ultra-fine entity tagging task was never introduced before the dedicated paragraph.

The task for which the model is tested does not reflect the paper's primary focus (shared propertied of diverse concepts). Such a discrepancy is probably due to missing research questions and main contributions in the introduction.

**Reproducibility:**

3: Could reproduce the results with some difficulty. The settings of parameters are underspecified or subjectively determined; the training/evaluation data are not widely available.

**Reviewer Confidence:**

3: Pretty sure, but there's a chance I missed something. Although I have a good feel for this area in general, I did not carefully check the paper's details, e.g., the math, experimental design, or novelty.

---

> ### Author Rebuttal · Authors · 2023-08-25
>
> We thank the reviewer for their careful reading of our paper. We are puzzled, however, about the discrepancy between the low scores and the minor nature of the stated “reasons to reject”.
>
> Note that we do not use ChatGPT directly. We have experimented with the use of ChatGPT for generating training data, but this step is not essential for our approach. Indeed, the model that was trained on ConceptNet performs comparably to the model that uses both ConceptNet and ChatGPT data (and better than the model that only uses the ChatGPT generated training data). Moreover, an inspection of the ChatGPT generated data reveals that it is highly accurate. In particular, we manually inspected the first 500 (concept,property) pairs and only found 6 errors, i.e. an error rate of 1.2%.
>
> The task of ultra-fine entity typing is explained in Section 3. We did not describe it in the introduction to avoid overlap (since this is a short paper) and because we wanted to focus the introduction on the key contribution of this paper, namely the idea of identifying shared properties as an alternative to learning embeddings.
>
> We will add a short paragraph to the introduction to make the research question and contributions more explicit, and to make the link with ultra-fine entity typing.

---

### Meta-Review · Area_Chair_9bHa · 2023-09-11

**Recommendation:** 5

**Metareview:**

The short paper deals with analysis of concepts collected in a large knowledge base. The method leverages LLMs to find common properties  for a given two concepts, which is opposed to a common practice of distilling concepts from texts. The paper presents a rare approach to concept representation, which is supported by solid experiments.
All three reviewers are generally agree in their comments and scores. However, some crucial points are left out from the main article body and presented either in appendixes or in rebuttal responses. The authors should think carefully how to use an extra page in the final version.

---

### Decision · Program_Chairs · 2023-10-07

**Decision:**

Accept-Main

**Comment:**

The short paper deals with analysis of concepts collected in a large knowledge base. The method leverages LLMs to find common properties  for a given two concepts, which is opposed to a common practice of distilling concepts from texts. The paper presents a rare approach to concept representation, which is supported by solid experiments.
All three reviewers are generally agree in their comments and scores. However, some crucial points are left out from the main article body and presented either in appendixes or in rebuttal responses. The authors should think carefully how to use an extra page in the final version.